# Dynamic Effects of Climate and Land Use Policies on Water Yield in Drylands—A Case Study in the Northwest of China

Li An [1,2], Shuai Zhong [1,2,3,*] and Lei Shen [1,2,3]

1. Institute of Geographic Sciences and Natural Resources Research, Chinese Academy of Sciences, Beijing 100101, China
2. College of Resources and Environment, University of Chinese Academy of Sciences, Beijing 100049, China
3. Key Laboratory of Carrying Capacity Assessment for Resource and Environment, Ministry of Natural Resources, Beijing 101149, China
* Correspondence: zhongshuai@igsnrr.ac.cn

**Abstract:** Water yield as an important ecosystem service for mitigating water scarcity in drylands, is quite sensitive to land use and climate change. Evaluating the response of water yield to land use policies in drylands under climate change is conducive to sustainable water resource management. Taking the Hohhot-Baotou-Ordous-Yulin region in the northwest of China as an example, this study used the methods of the InVEST model, trend analysis, comparative experiment, PLUS model to explore the spatial-temporal trends and driving factors of water yield variation, as well as to simulate the future water yield under different land use policy scenarios. The results showed that (1) water yield in the study area fluctuated and increased from 14.14 mm to 46.59 mm during 2000 to 2020, the places with a significant increasing trend of water yield were mainly distributed in the southeast loess hilly area; (2) climate change is the major driving factor of water yield variation with a contribution rate of 85.8–99.6%, but land use change played an increasingly important role in affecting water yield; (3) the Grain-Security-Dominated (GSD) land use policy scenario would generate the highest water yield in 2030 under climate conditions of SSP1-2.6, SSP2-4.5, SSP3-7.0, SSP5-8.5, while the Regional-Greening-Dominated (RGD) land use policy scenario has the lowest water yield in the future. The results could provide reference for the decision-making process of ecological restoration and land use planning in drylands.

**Keywords:** water yield; climate change; land use policy; ecosystem service; scenario analysis; dryland region

## 1. Introduction

Ecosystems provide a range of welfare and benefits called ecosystem services (ESs), which are of fundamental importance to human well-being, health, livelihoods, and survival [1,2]. Water yield, generally defined as the amount of precipitation minus actual evapotranspiration [3,4], is a water-related ecosystem service that determines the available water for human consumption [5,6]. It indicates the quantity of water resources generated from the hydrological cycles of ecosystems; thus, it has become a focal point in the cross-disciplinary fields of hydrology, ecology, and water resource management [7].

Water yield is especially critical in drylands, which account for over 40% of the terrestrial land surface [8], and have been proved to be expanding under the influence of climate change [9,10]. Drylands are not only faced with conspicuous shortage of water [11,12], but also more affected by climate change than humid lands [13]. Under the background of climate change, land use patterns also play a vital role in affecting hydrological conditions [14,15]. Evaluating the effect of climate and land use change on dryland water yield is helpful to put forward corresponding policies for improving dryland water yield service and realizing water-related sustainable development goals [16–18]. In this respect, cases in the northwest of China could provide valuable insights because these regions experienced

great land use transformations under land use policies, such as the Three-North Shelter Forest program and the Grain-For-Green (GFG) program [19].

As a typical dryland region, the Hohhot-Baotou-Ordous-Yulin (HBOY) region in the northwest of China is ecologically fragile and ESs are sensitive to climate and land use change. Since 2000, the GFG program has been implemented in this area to improve the vegetation coverage, leading to the transformation from cropland to forest and grassland [20]. Meanwhile, a rapid urbanization was witnessed in this area with the expansion of built-up land. However, in recent years grain security is gaining more and more attention, and retaining cropland area is an important policy goal in the HBOY region according to the National Land Planning Outline towards 2030, which intensifies the land competition between reforestation and cropland protection. Therefore, the motivation of this study is to: (1) clarify the spatial-temporal patterns of water yield in the HBOY region and its dynamic response to climate and land use change during 2000 to 2020; (2) explore the effect of different land use policies on water yield in 2030, in order to propose the corresponding policy implications and realize more sustainable water resource management.

*Progress in Assessing Water Yield and the Literature Gap*

With the increasing awareness of the environmental consequences and risks [21,22] brought by global warming, it has become a hot topic to investigate the response of water yield to climate change. Climatic factors such as precipitation have a fundamental influence on hydrological balance, thus affecting confluence, runoff, and water-related ESs [23,24]. Xu et al. [25] quantified the effect of precipitation, temperature, solar radiation and vegetation coverage on water yield in the Qilian Mountains, and revealed the risk of diminished water yield in mid-elevation dryland mountains under climate change. Since land use patterns have a nonnegligible impact on hydrological conditions [26], a large body of literature elaborated and compared the contribution of climate and land use change to water yield variation [27,28]. For example, Clerici et al. [29] studied the cumulative effect of land use and climate change on water yield in two peri-urban areas of Colombia, and the results showed that climate change scenarios had a greater impact on water yield than land use change scenarios. Lang et al. [30] found that the impact of rainfall change on water yield was up to 97.44% in the Sancha River Basin, while the contribution rate of land use change was 2.56%. Although a straight-forward conclusion could be derived from previous literature that climate change contributed dominantly to the variation of water yield compared to land use change, the calculation results of contribution rates may vary greatly in different years given that climatic factors is highly unstable [31]. Identifying the dynamic features in contribution rates could help to answer a new question whether the relative effect of climate and land use change on water yield is on the rise or is decreasing.

Under the background of climate change, a series of policy tools are put forward for realizing sustainable water resource management. Most of these policies aim at improving water use efficiency, water quality/quantity, or regulating the water market to realize a fair allocation of water resources among water users [32–35]. Land use policies have a fundamental influence on land use patterns, thus affecting water yield and available water resources [36]. Ke et al. [37] simulated the impact of cropland protection on water yield in Wuhan city, and found that a strict cropland protection policy would bring about a higher water yield. Ferreira et al. [38] examined the effect of deforestation/reforestation on water-related ecosystem services in a Brazilian megalopolis under climate change. The results showed that deforestation scenario would generally lead to the decrease in water quality and increase in water quantity, while reforestation would improve water quality and water quantity in urbanized areas. Scenario analysis in above studies played a vital role in informing decision-makers the influence of interested land use policy on water yield. However, few examples of literature considered the effect of land use policies in dryland regions under different climate change scenarios.

There are many approaches for assessing water yield, among which measured water yield based on actual observed data has high accuracy, yet it is limited in illustrating the

spatial distribution features of water yield. Therefore, modeling tools based on remote sensing data play an important role in assessing water yield. With the rapid development of remote sensing and geographical information technologies [39], several modelling tools such as soil and water assessment tool (SWAT) [40], artificial intelligence for ecosystem services (ARIES) [41], integrated valuation of ecosystem services and tradeoffs (InVEST) [42] have been established and applied in the assessment of water yield. Among them, the InVEST model has two outstanding advantages: firstly, most input and output of the InVEST model are in the format of raster data, which provides spatially explicit information for calculating water yield on different land use types, and makes it viable to estimate water yield under land use change scenarios [43]; secondly, the InVEST model is one of the most widely used models for assessing water yield with mature calibration approaches [44], which guarantees reliability and robustness in the results [45].

Therefore, this study applied the InVEST model and several auxiliary methods to assess water yield in the HBOY region. Two contributions are expected in this study to previous literature: (1) make up for the knowledge gap on how the effect of climate and land use change on water yield varies dynamically in different years; (2) investigate the effect of different land use policies on dryland water yield in the future, with consideration of climate change scenarios in shared socio-economic pathways (SSPs).

## 2. Materials and Methods

### 2.1. Study Area

The HBOY region (36°48′50″ N–42°44′5″ N, 106°28′16″ E–122°18′7″ E), covering an area of $17.46 \times 10^4$ km$^2$, is composed of four prefecture-level cities: Hohhot, Baotou, Ordous, Yulin in the northwest of China (Figure 1). The highest point in the HBOY region is 2338 m in the Daqing mountain and the lowest point is 561 m. The average annual temperature is 7.8 °C. With a temperate continental monsoon climate, the region has an average annual precipitation of 320 mm and monthly mean precipitation ranging from 3 mm to 129 mm. As a proposed urban agglomeration in the national urbanization plan, this region experienced a rapid urbanization process from 2000 to 2020. The area of construction land increased from 871.64 km$^2$ to 2329.72 km$^2$, and the non-agricultural population in the region increased from 2.83 million to 10.1 million, leading to a larger water demand for urban life and industrial production. Enhancing water yield is critical for mitigating water scarcity and realizing sustainable development in this region.

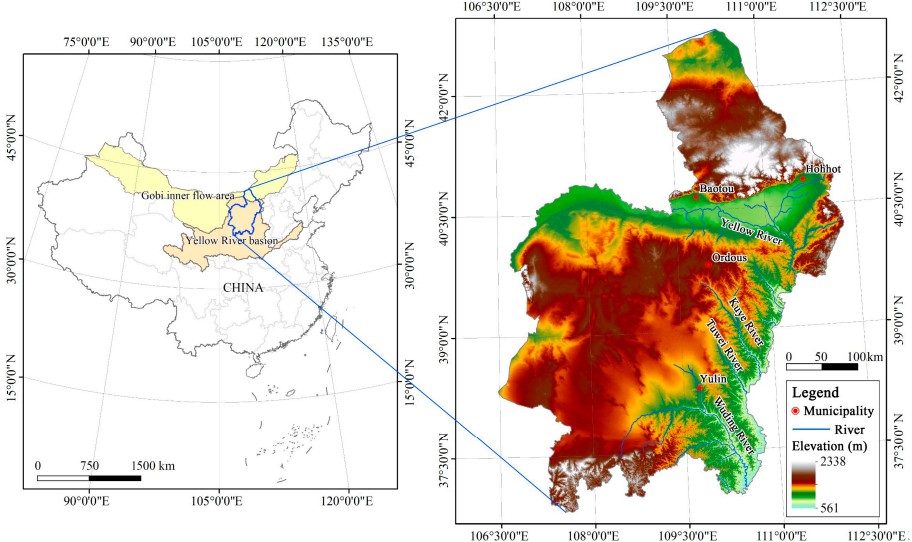

**Figure 1.** Location and digital elevation model of the HBOY region.

### 2.2. Methods

The methods in this study include the InVEST model, PLUS model, trend analysis and comparative experiments. Using the annual water yield module of the InVEST model, the spatial-explicit water yield from 2000 to 2020 was evaluated, on this basis the variation trends of water yield were identified on the raster grid scale. By means of comparative experiments, the yearly effects of climate change and land use change on water yield, as well as their relative contribution rates were quantified. The PLUS model was applied to simulate the land use layouts in 2030 in the study area, and the future water yield was projected under various land use policy and climate change scenarios. A workflow of the methods used in this study is given in Figure 2.

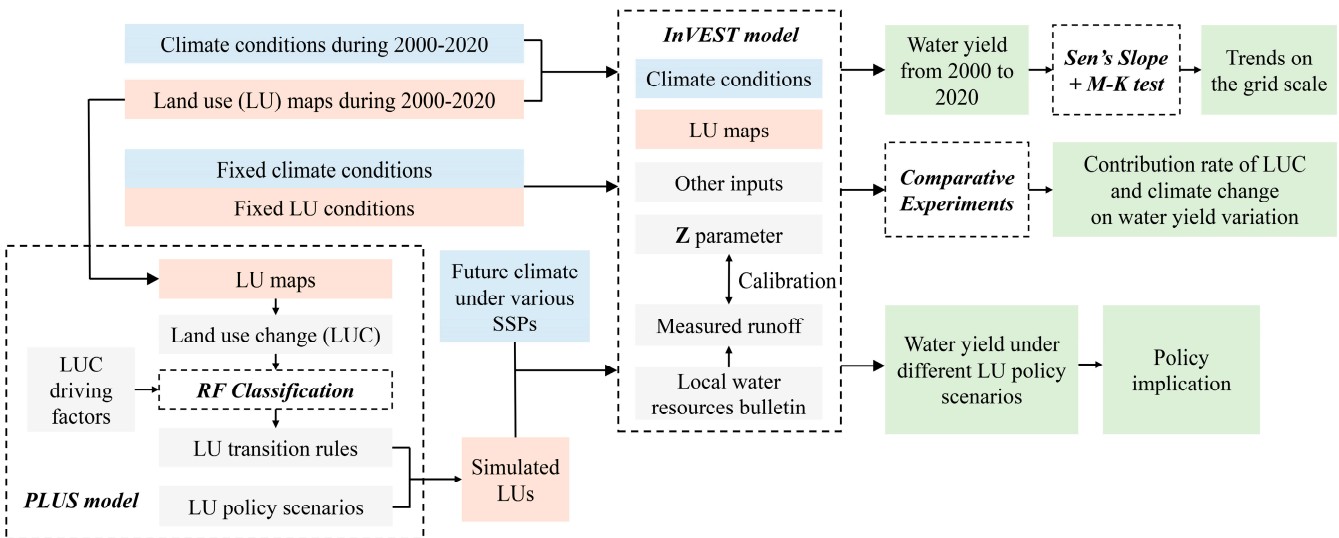

**Figure 2.** Work flow of the study.

#### 2.2.1. InVEST Model

The annual water yield module of the InVEST model is based on the Budyko curve and annual average precipitation [42], which assumes a simplified confluence process including surface runoff, soil runoff, and base flow [46]. The sum and average of water yield are calculated at sub-watershed level according to the principle of water balance. The annual water yield of each pixel is calculated in accordance with the difference between precipitation and evapotranspiration [37], which can be expressed as Equation (1):

$$Y_{ij} = \left(1 - \frac{AET_{ij}}{P_{ij}}\right) \times P_{ij} \tag{1}$$

where $Y_{ij}$ is the water yield on pixel *i* for land use type *j* (mm); $AET_{ij}$ is the actual evapotranspiration on pixel *i* for land use type *j* (mm); $P_{ij}$ is the precipitation on pixel *i* for land use type *j* (mm).

For vegetated land, *AET* is calculated as Equation (2):

$$\frac{AET_{ij}}{P_{ij}} = 1 + \frac{PET_{ij}}{P_{ij}} - \left[1 + \left(\frac{PET_{ij}}{P_{ij}}\right)^{\omega_{ij}}\right]^{\frac{1}{\omega_{ij}}} \tag{2}$$

where $PET_{ij}$ is the potential evapotranspiration on pixel *i* for land use type *j* (mm); $\omega_{ij}$ is an empirical fitting parameter that characterizes the land surface properties of catchments [44], which can be calculated by Equations (3) and (4) [47]:

$$\omega_{ij} = Z \frac{AWC_{ij}}{P_{ij}} + 1.25 \tag{3}$$

$$AWC_{ij} = \min\left(ld_{ij}, rd_{ij}\right) \times PAWC_{ij} \tag{4}$$

where $AWC_{ij}$ is the volumetric (mm) plant available water content; $ld_{ij}$ is the depth of root restricted layer, often given as the depth at which 95% of a vegetation type's root biomass occurs; $rd_{ij}$ is the vegetation rooting depth; $PAWC_{ij}$ is the plant available water capacity; $Z$ is an empirical constant that captures the local precipitation pattern and additional hydrogeological characteristics.

### 2.2.2. Trend Analysis

The trend of water yield on each pixel was detected by means of Sen's slope method [48] and Mann-Kendall (M-K) test [49]. Firstly, the nonparametric Sen's slope method is applied to quantify the magnitude of trends in water yield time series. A negative Q value in Sen's slope method indicates a downward (decreasing) trend and a positive Q value indicates an upward trend. Then, the significance level of trends in water yield is identified by Mann-Kendall test, which is widely used for detecting trends in hydrological parameters [50] because of its robustness for non-normally distributed data and outliers. On this basis, the trend of water yield on each pixel and its significance level was visualized by ArcGIS mapping tools.

### 2.2.3. Comparative Experiment

The effect of land use and climate change on water yield could be quantified by simulating different land use and climate conditions for comparative experiments [30,31]. In order to calculate the dynamic contribution rates of land use and climate change in each year, we took 2000 as the base year and established two hypothetical conditions: one is the yearly climate in 2000–2020 along with fixed land use in 2000, and the other is the yearly land use in 2000–2020 along with fixed climate in 2000. The contribution of land use and climate change was derived by comparing the actual water yield with water yield under hypothetical conditions (Equations (5) to (8)).

$$E_{CC} = WY_{CC\&LUC} - WY_{fixed\_CC} \tag{5}$$

$$E_{LUC} = WY_{CC\&LUC} - WY_{fixed\_LUC} \tag{6}$$

$$CR_{CC} = \frac{|E_{CC}|}{|E_{CC}| + |E_{LUC}|} \times 100\% \tag{7}$$

$$CR_{LUC} = \frac{|E_{LUC}|}{|E_{CC}| + |E_{LUC}|} \times 100\% \tag{8}$$

where $E_{CC}$ is the change in water yield relative to the base year due to climate change, $E_{LUC}$ is the change in water yield relative to the base year due to land use change, $WY_{CC\&LUC}$ is the water yield under actual climate and land use conditions, $WY_{fixed\_CC}$ is the water yield under actual land use and fixed climate conditions, $WY_{fixed\_LUC}$ is the water yield under actual climate and fixed land use conditions. $CR_{CC}$ and $CR_{LUC}$ are the contribution rates of climate change and land use change on water yield change, respectively. The $E_{CC}$ and $E_{LUC}$ were taken as absolute value when calculating contribution rates to make sure $CR_{CC}$ and $CR_{LUC}$ ranging from 0 to 100%.

### 2.2.4. Patch-Generating Land Use Simulation (PLUS) Model

The cellular automata (CA) model has been proved to be one of the most reliable tools for land use layout simulation because of its robustness and flexibility. As a recent CA-based model, PLUS is the coupling of rule-mining framework based on land expansion analysis strategy (LEAS) and cellular automata CA model based on multi-type random patch

seeds [51]. LEAS overcomes the disadvantages of transition analysis strategy (TAS) [52] and pattern analysis strategy (PAS) [53] by extracting the changing part of two periods of land use data, and adopting the random forest classification (RFC) algorithm to obtain the land use transition rules. The algorithm can be expressed as Equation (9):

$$P_{i,j}^d(x) = \frac{\sum_{n=1}^{M} I(h_n(x) = d)}{M} \tag{9}$$

where, $P_{i,j}^d(x)$ denotes the development potential of land use type *j* at cell *i*; M is the total number of decision trees; $I(\cdot)$ is the indicative function of the decision tree set; *x* is a vector composed of multiple driving factors; $h_n(x)$ represents the forecasting type of the *n*-th decision tree for vector *x*; *d* is either 0 or 1, indicating if there were other land use types that transformed to land use type *j*. The driving factors used for forecasting land expansion potential include social-economic factors and natural factors. Six social-economic factors and six natural factors were selected for uncovering the land expansion rules and simulating the land use layout in 2030, as is shown in Table 1.

**Table 1.** Driving factors for land use change simulation.

| Category | Factor name | Category | Factor Name |
|---|---|---|---|
| Social-economic factor | GDP density | Natural factor | Accessibility to water land |
| | Population density | | Soil type |
| | Accessibility to municipality | | Elevation |
| | Accessibility to primary road | | Slope |
| | Accessibility to highway | | Annual average precipitation |
| | Accessibility to railway | | Annual average temperature |

After obtaining the land use transition rules by LEAS in the study area, CA model based on multi-type random patch seeds was used to simulate the land use in 2030. Here we proposed three land use policy scenarios: the first is Natural Development (ND) scenario, which assumes that the future land use policies remain the same with the existing ones, and the expansion trend of each land use type such as urban expansion and GFG program continues in the future; the second is Regional-Greening-Dominated (RGD) scenario, simulating the policy preference for vegetation protection and a more vegetated land use layout; the third is Grain-Security-Dominated (GSD) scenario, assuming that the GFG program is called off in the future for grain security. The methods for predicting the area of different land use types in the proposed land use policy scenarios are described in Table 2.

**Table 2.** Settings for various land use policy scenarios.

| Land Use Policy Scenarios | Descriptions |
|---|---|
| Natural Development (ND) scenario | The area of various land types is predicted by Markov-chain method, assuming that the transition probability between various land use types from 2020 to 2030 is the same as that from 2010 to 2020. |
| Regional-Greening-Dominated (RGD) scenario | The area of various land types is predicted by adjusted Markov matrix in which the transition from forest and grassland to other lands is forbidden. |
| Grain-Security-Dominated (GSD) scenario | The area of various land types is predicted by adjusted Markov matrix in which the transition from cropland to forest or grassland is forbidden while the transition probability from forest and grassland to farmland remains the same as that in the original matrix. |

### 2.3. Data Sources and Preprocessing

There are seven types of data required for water yield calculation and land use simulation, including meteorological data, soil data, land use data, road data, topographic data,

hydrologic data, and social-economic data. The data sources are summarized in Table 3. The input needed for the InVEST model and PLUS model are directly from the these data or derived via data preprocessing based on them. The input raster data for the InVEST model and PLUS model was resampled to 1 km resolution, and the grid-scale analysis of water yield was also carried out on 1 km resolution.

**Table 3.** Data sources.

| Data type | Data description | Data source | Data usage |
|---|---|---|---|
| Meteorological data | Precipitation data from 2000 to 2020 Temperature data from 2000 to 2020 | China Meteorological Science Data Center (http://data.cma.cn/) The accessed date: 17 December 2021 | Calculation of reference evapotranspiration and water yield |
| | Extraterrestrial radiation | The United Nations Food and Agriculture Organization (FAO) Irrigation Drainage Paper [54] | |
| | Future precipitation and temperature | Coupled Model Intercomparison Project (Phase 6) (https://esgf-node.llnl.gov/projects/cmip6/) The accessed date: 26 March 2022 | |
| Soil data | Soil reference depth, clay, sand, silt and organic matter contents, in China soil map based on harmonized world soil database (HWSD) (v1.1) | National cryosphere desert data center (https://www.ncdc.ac.cn) The accessed date: 16 August 2021 | Calculation of plant available water capacity and water yield |
| Hydrologic data | Measured surface runoff and water resource quantity, in the water resources bulletin of Hohhot, Baotou, Ordous and Yulin | Municipal water conservancy bureau of Hohhot, Baotou, Ordous and Yulin The accessed date: 21 March 2022 | Calibration of Z parameter in InVEST model |
| Land use data | The 30 m annual land cover datasets and its dynamics in China from 1990 to 2020 | Earth System Science Data (https://essd.copernicus.org) The accessed date: 5 April 2022 | Calculation of water yield; rule-mining of land expansion |
| Road data | Highway, railway, primary road, etc | OpenstreetMap (https://www.openstreetmap.org) The accessed date: 13 July 2021 | Rule-mining of land expansion |
| Topographic data | Digital elevation model (DEM) | Resource and Environment Sciences and Data Center, Chinese Academy of Science (https://www.resdc.cn) The accessed date: 18 January 2022 | Rule-mining of land expansion |
| Socio-economic data | GDP density raster Population density raster Location of municipalities | Resource and Environment Sciences and Data Center, Chinese Academy of Science (https://www.resdc.cn) The accessed date: 18 January 2022 | Rule-mining of land expansion |

### 2.3.1. Data Preprocessing for Water Yield Calculation

The input data for the InVEST model include annual precipitation, reference evapotranspiration, land use map, soil reference depth, PAWC, the layer of watershed, biophysical table and Z parameter. Reference evapotranspiration is referred to as potential evapotranspiration under special constraints of plant and surface conditions [55]. The best approach for estimating reference evapotranspiration is Penman-Monteith equation since it can be used globally without local calibration [56]. The drawback of this method is lack of the actual measured input data in requirement. To address this problem, the empirical Hargreaves equation [57] was recommended by FAO as an alternative method, which generates superior results than the Penman-Monteith model when there is less available data. Therefore, the Hargreaves equation was used to calculate the reference evapotranspiration (Equation (10)):

$$ET_0 = 0.0013 \times 0.408 \times RA \times (T_{av} + 17) \times (TD - 0.0123P)^{0.76} \tag{10}$$

where $ET_0$ is reference evapotranspiration (mm), $RA$ is the extraterrestrial radiation (MJ·m$^{-2}$·d$^{-1}$), $T_{av}$ is the average of the mean daily maximum and mean daily minimum

temperatures for each month (°C), $TD$ is the difference between mean daily maximum and mean daily minimums for each month (°C), $P$ is the monthly precipitation (mm). In previous studies, the parameters in the Hargreaves equation were proved to have good serviceability in the loess plateau and inner Mongolia [58,59], and thus are applicable in the study area.

$PAWC$ is the difference between field capacity and wilting point, ranging from 0 to 1. The calculation function for $PAWC$ is as Equation (11) [60]:

$$PAWC = 54.509 - 0.132 \times sand - 0.003 \times sand^2 - 0.055 \times silt - 0.006 \times silt^2 - 0.738 \times clay + 0.007 \times clay^2 - 2.688 \times OM + 0.501 \times OM^2 \tag{11}$$

where $clay$, $sand$, $silt$, $OM$ are percentage proportion of clay, sand, silt and organic matter in the soil, respectively.

Based on DEM data, the watershed shapefile layer is obtained using hydrological analysis tools in ArcGIS 10.8. The parameters needed in the biophysical table are determined according to previous studies [46,61] and FAO guidelines [54]. Z parameter is calibrated as 6.4 by comparing the calculated water yield amount by the InVEST model and the measured surface runoff in local water resources bulletin.

### 2.3.2. Data Preprocessing for Land Use Simulation

Accessibility to municipality, railway, highway, primary road and water land are Euclidian distances derived from ArcGIS 10.8 spatial analyst tool. The raster layer of slope was obtained from DEM data using 3D analysis tool in ArcGIS 10.8.

## 3. Results

### 3.1. Changes in Climate and Land Use Patterns

A warming-and-humidifying climatic feature in the HBOY region was shown from the temporal change of annual precipitation and mean temperature, which are two important climatic drivers of water yield [62]. Figure 3 depicted the annual precipitation, mean temperature, AET and PET in the HBOY region from 2000 to 2020. The average annual precipitation and mean temperature were 354.7 mm and 7.65 °C respectively, with changing rates of 2.7 mm·year$^{-1}$ and 0.021 °C·year$^{-1}$. The highest annual precipitation was observed in 2012 (438.68 mm) and the lowest annual precipitation was in 2005 (6.03 mm). With increases in both annual precipitation and mean temperature in the study area, the PET had significant increasing trends ($p < 0.05$). There is also an increasing trend of the AET ($p = 0.091$) at a rate of 1.88 mm year$^{-1}$. The highest AET was in 2018 (340.46 mm), and the lowest AET occurred in 2005 (211.72 mm) along with the lowest precipitation. The average value of AET was 292.3 mm, indicating that on average 82.4% of the precipitation evaporated into the atmosphere.

Under the influence of urbanization process and ecological measures such as the GFG project and vegetation restoration, the land use layout in the HBOY region has changed greatly in the past 20 years (Table 4), dominantly characterized by the shrinkage of cropland and barren land, and the expansion of grassland, forest, and built-up land. From 2000 to 2020, the area of cropland decreased from 31,128.95 km$^2$ to 26,511.06 km$^2$, mostly transformed into grassland and built-up land. There was also a net conversion of barren land to grassland. As a result, the area of grassland and built-up land increased by 9.90% and 167.24% from 2000 to 2020, respectively. The places where cropland transformed to grassland were mainly located in the northern Daqing Mountain and the southeast loess hills; the transformation of barren land to grassland primarily happened in the middle and northwest of the HBOY area; and the expansion of built-up land was around the central urban area of Hohhot and Baotou.

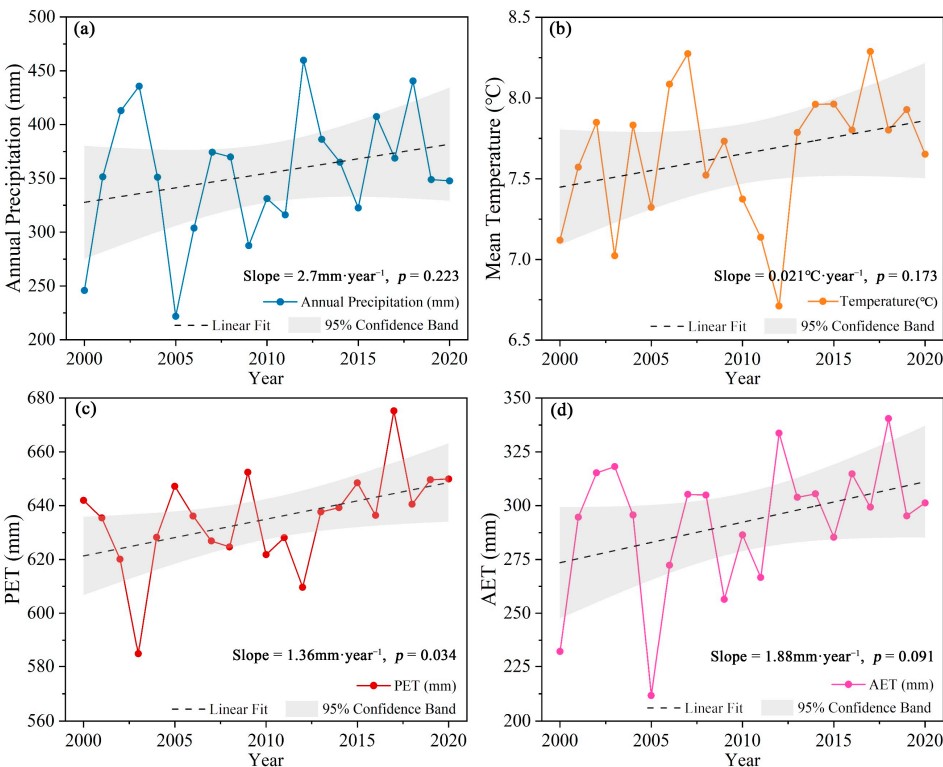

**Figure 3.** Annual precipitation (**a**), mean temperature (**b**), PET (**c**), AET (**d**) in the HBOY region from 2000 to 2020.

**Table 4.** Transition matrix of land use types between 2000 and 2020 in the HBOY region (Unit: km$^2$).

| Land Use Type | | 2000 | | | | | | |
|---|---|---|---|---|---|---|---|---|
| | | Cropland | Forest | Grassland | Water | Barren | Built-Up Land | Other Land |
| 2020 | Cropland | 14,807.92 | 8.74 | 10,865.22 | 124.84 | 522.86 | 179.33 | 2.15 |
| | Forest | 38.28 | 376.32 | 623.11 | 0.72 | 1.17 | 1.44 | 7.98 |
| | Grassland | 15,036.61 | 441.80 | 104,141.73 | 129.43 | 11,522.93 | 300.39 | 7.12 |
| | Water | 185.10 | 1.26 | 197.17 | 270.84 | 65.30 | 15.76 | 0.31 |
| | Barren | 81.69 | 0.90 | 2923.99 | 22.43 | 9061.72 | 10.36 | 0.00 |
| | Built-up land | 979.35 | 1.17 | 975.74 | 57.83 | 129.52 | 471.16 | 0.28 |
| | Other land | 0.00 | 3.51 | 4.05 | 0.00 | 0.09 | 0.00 | 0.36 |

### 3.2. Spatial-Temporal Variation of Water Yield

For temporal trends, the changing rate of annual water yield was 0.82 mm·year$^{-1}$ ($p$ = 0.482) from 2000 to 2020, with an average annual value of 62.5 mm. The highest value of annual water yield was 126.2 mm in 2012, and the lowest water yield occurred in 2005 (6.03 mm). On the grid scale, about 6% of the grids showed a significant increasing or decreasing trend ($p$ < 0.1) of water yield, while there was no significant trend in most places (Figure 4). The grids with increasing trends were distributed in the eastern hilly area of Yulin and the northwest of Ordous, which accounted for 5.8% of the study area. There is a small region in the middle east where the water yield had significant downward trends, accounting for 0.2% of the study area.

The distribution of water yield in the HBOY region showed distinct spatial heterogeneity. Under the influence of East Asia monsoon [63], the southeast of the study area has more seasonal rainfall and more evapotranspiration. As a result, the water yield was higher in the southeast than that in the northwest, similar to the spatial distribution of precipitation and AET, as is shown in Figure 5. Raster girds with high values (>100 mm) of average annual water yield accounted for 22.9% of the study area, mainly distributed in

the southeast part; while low values (<100 mm) of average annual water yield occurred in 77.1% the study area, generally in the middle and northwest proportion.

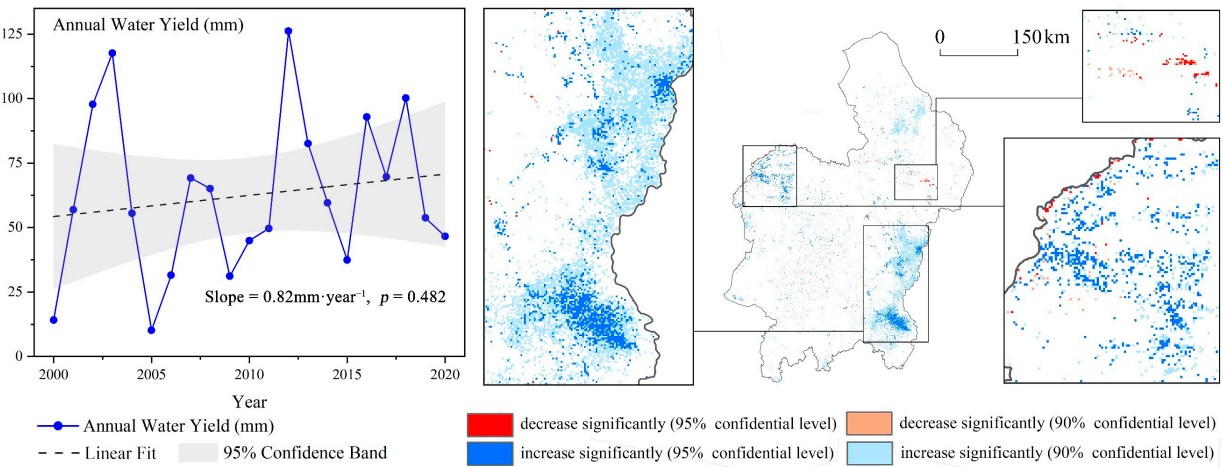

**Figure 4.** Spatial-temporal variation trends of water yield.

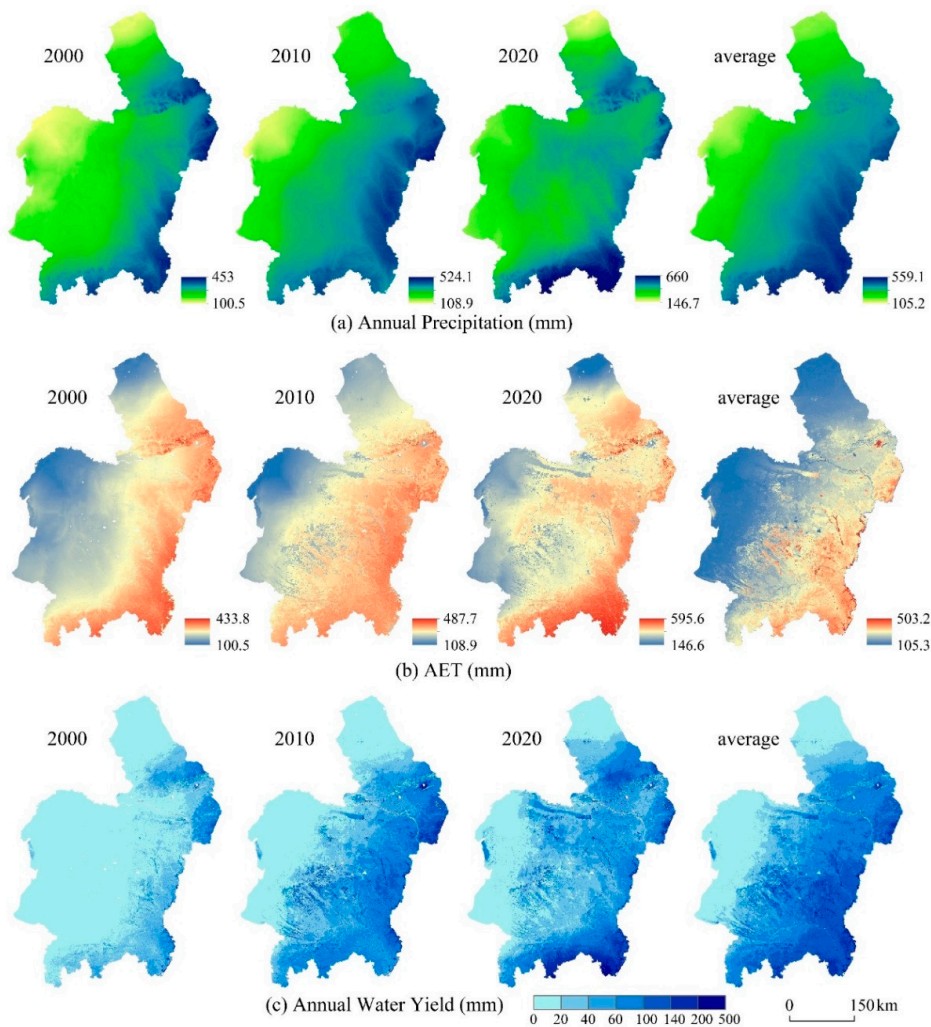

**Figure 5.** Spatial distribution annual precipitation (**a**), AET (**b**), and annual water yield (**c**) in the HBOY region.

### 3.3. Effects of Climate and Land Use Change on Water Yield

3.3.1. Effect of Climate Change

Climate change played a dominant role in the interannual variation of water yield in the HBOY region. The results of comparative experiments showed that the effect of climate change on water yield from 2000 to 2020 turns out to be promotion, as is shown in Figure 6a. The contribution rate of climate change to water yield variation ranged from 85.8% to 99.6%, with the highest rate in 2001 and the lowest in 2015. A noteworthy result is that despite climate change having a fundamental impact on the variation of water yield, there is a decreasing trend in the contribution rate at a rate of 0.373 percentage point per year.

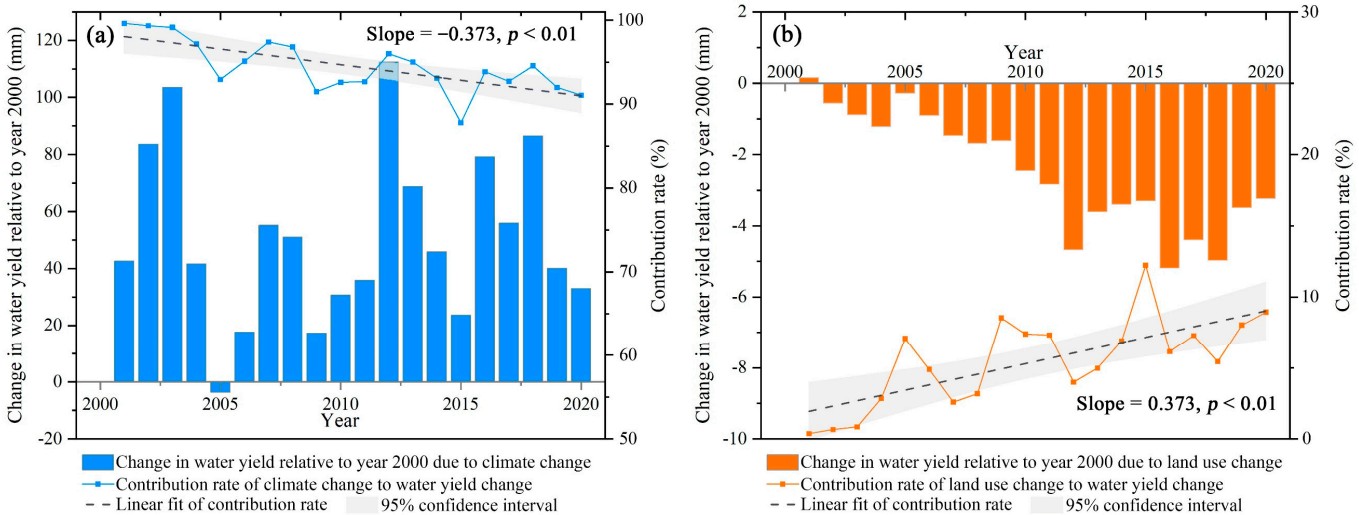

**Figure 6.** Contribution of climate change (**a**) and land use change (**b**) to water yield variation in the HBOY Region.

The effect of precipitation and temperature change on water yield change was further explored on the grid scale through Spearman correlation analysis (Figure 7). The results showed that the relationship between annual water yield and annual precipitation was a strong significant positive correlation (correlation coefficient $\geq$ 0.6, $p < 0.05$) in 97.2% of the grids and a middle/weak significant positive correlation (correlation coefficient is 0–0.6, $p < 0.05$) in 1.5% of the grids. Since precipitation directly affects the water input of ecosystems, the more the quantity of rainfall, the more the water yield amount. The relationship between annual water yield and annual mean temperature was a significant negative correlation (correlation coefficient < 0, $p < 0.05$) in 5.9% of the study area, while grids with no significant positive correlation between mean temperature and water yield ($p \geq 0.05$) accounted for 94.0% of the study area.

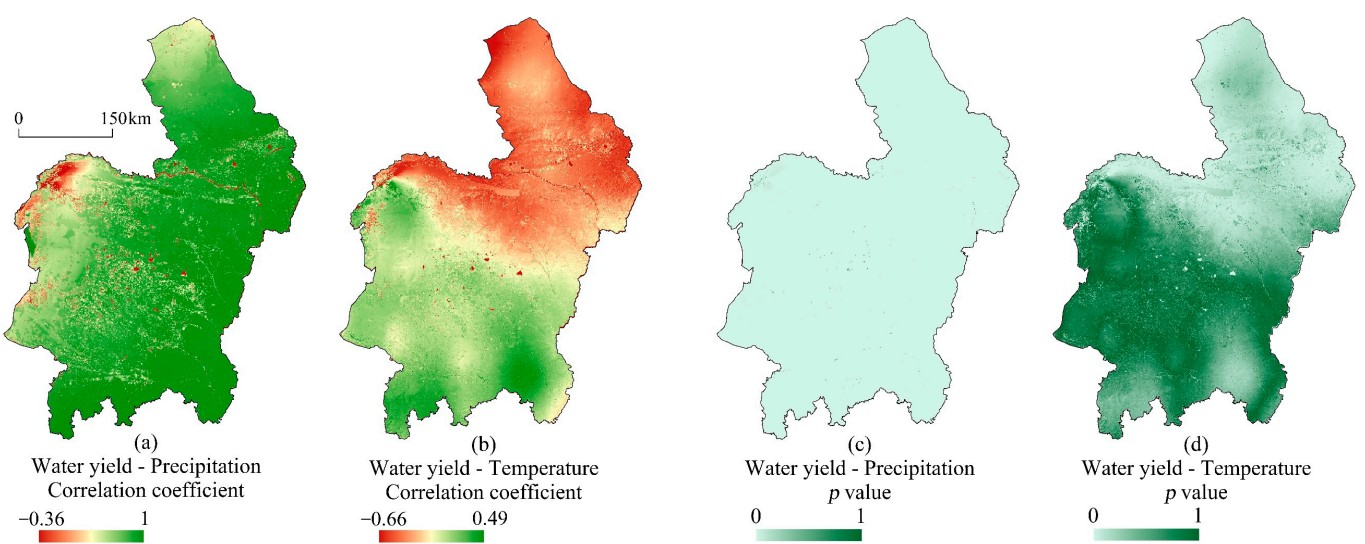

**Figure 7.** Correlation coefficient of precipitation and temperature with water yield (**a,b**) and the significance level (**c,d**) in the HBOY Region.

### 3.3.2. Effect of Land Use Change

The effect of land use change on water yield was mainly shown as negative from 2000 to 2020 (Figure 6b). Compared to the simulated water yield with fixed land use patterns, the water yield with actual land use change was 0.28 mm to 5.18 mm lower, except for 2001 in which land use change slightly elevated annual water yield by 0.16 mm. The average water yield varies greatly among different land use types, as is shown in Figure 8. The built-up area has the highest average water yield of 165.1 mm, followed by barren land (108.6 mm) and cropland (85.7 mm), while the average water yield of grassland (55.5 mm) and forest (27.7 mm) is relatively low. This result is similar to the observations reported by Lang et al. [30] and Ke et al. [37]. A cause of this phenomenon is that the more vegetated land use types such as grassland and forest needed more water for the bio-activity of plants and had more extensive transpiration [64], leading to more proportion of precipitation back into the atmosphere. The contribution rate of land use change has increased significantly during the past 20 years, signifying that the anthropic factor of land use change is playing a more and more important role in affecting water yield.

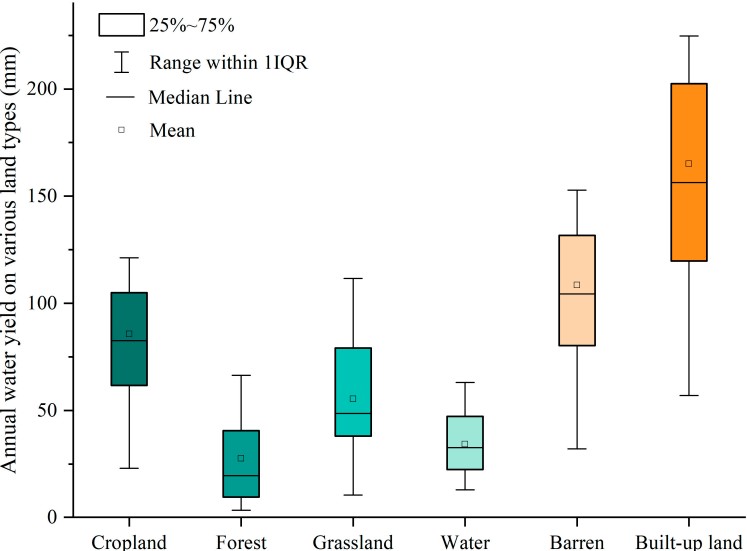

**Figure 8.** Water yield of different land use types in the HBOY Region.

### 3.4. Water Yield Simulation under Different Land Use Policy Scenarios

In order to explore the water yield under different land use policies, we simulated the land use patterns in various scenarios using PLUS model. Figure 9 showed the land use patterns in the HBOY region in 2020 and simulated ND, RGD, GSD scenarios. Here we took three key areas as examples to visually exhibit the difference between them. The first is the central urban area of Baotou in the north, the second is the Maowusu sandy area in the middle west, and the third is the loess hilly area in the southeast. It could be seen that there is less fragmented cropland in the RGD scenario than that in the ND scenario, especially in the southern hilly area and north of Daqing mountain, while there is more grassland converted to cropland in the GSD scenario. In simulated ND, RGD and GSD scenarios, the built-up land expands around the central urban areas of Hohhot, Baotou and Yulin, which is in accordance with the urbanization process in the HBOY region.

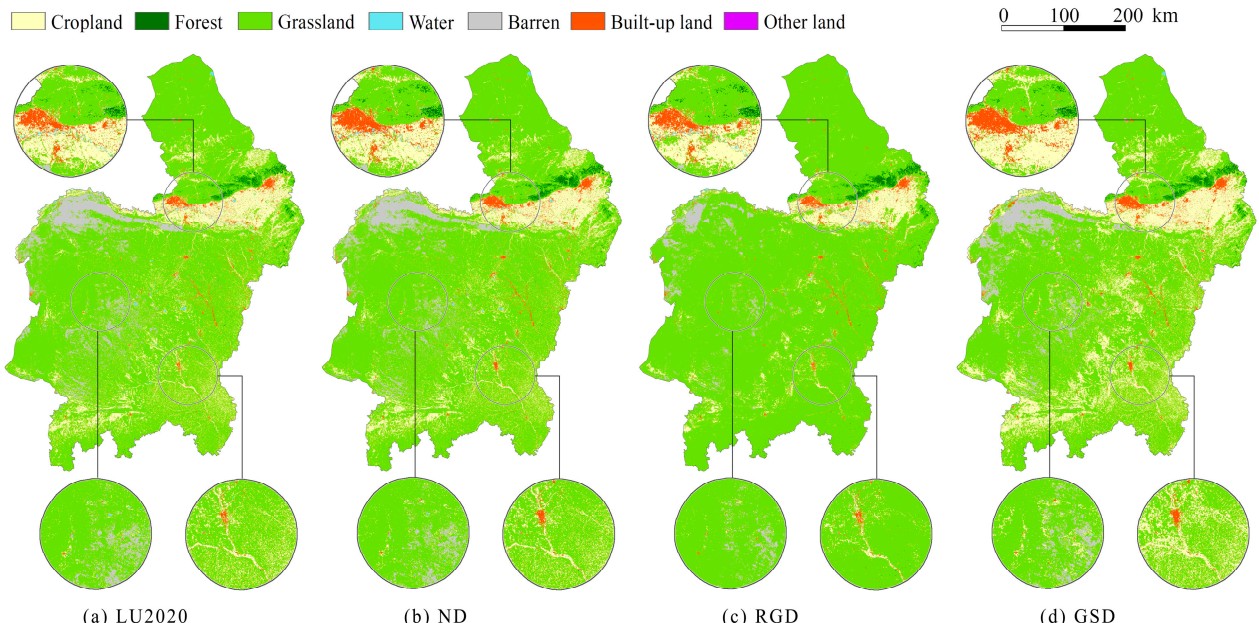

**Figure 9.** Land use patterns in 2020 (**a**), ND scenario (**b**), RGD scenario (**c**) and GSD scenario (**d**).

Figure 10 showed the simulated water yield under different land use policy scenarios and climate conditions. Four climate conditions in the future are considered [65,66]: SSP1-2.6 represents a pathway in which global warming is well addressed, with low radiative forcing and greenhouse gas (GHG) emissions; SSP2-4.5 represents a middle road of climate change in which global warming is moderate; SSP3-7.0 represents a pathway with medium-high radiative forcing and steadily intensified global warming; SSP5-8.5 represents a climate change scenario under high radiative forcing and GHG emissions. The simulation results show that the GSD scenario has the highest water yield in all future climate conditions, most likely due to decreased evapotranspiration from less vegetation coverage. In contrast, the RGD scenario has a lower water yield than that with 2020 land use patterns in SSP1-2.6, SSP2-4.5, SSP3-7.0, SSP5-8.5 by 1.14 mm, 2.08 mm, 1.42 mm, 3.20 mm respectively. The water yield in the ND scenario is similar to that with 2020 land use patterns, slightly higher in SSP1-2.6 and SSP3-7.0 by 0.05 mm and 0.04 mm, lower in SSP2-4.5 and SSP5-8.5 by 0.05 mm and 0.21 mm. The simulation results also reflected that there's distinctly greater difference in water yields under various climate conditions than water yields under various land use patterns. This result is consistent with the conclusions above that climate change is the major driving factor of water yield.

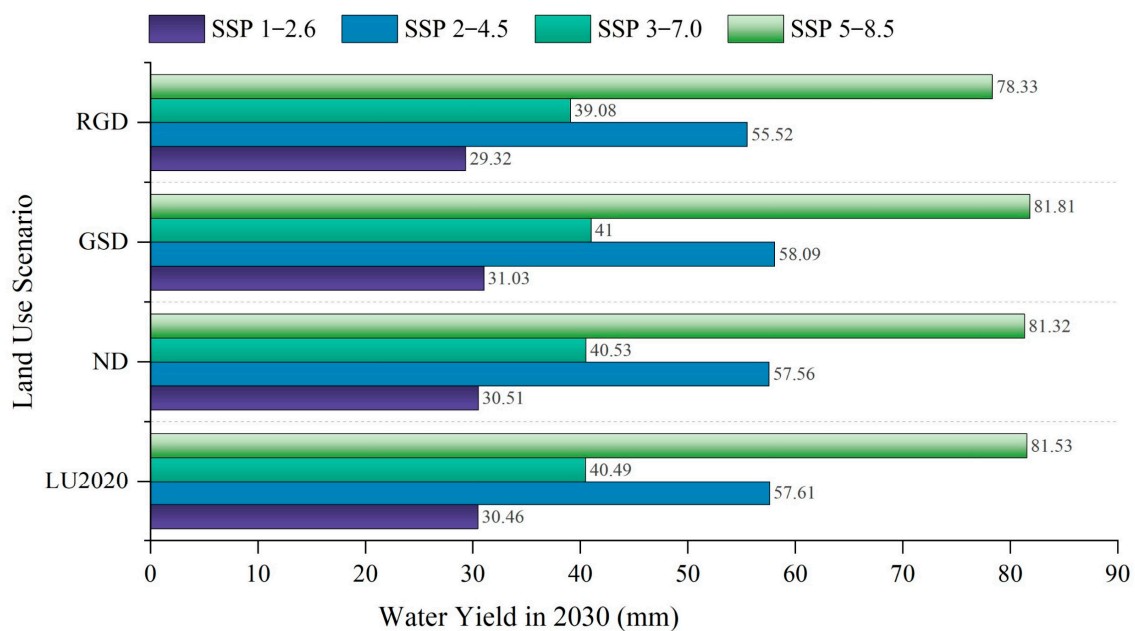

**Figure 10.** Water yield in 2030 in different land use policy and climate change scenarios.

## 4. Discussion

### 4.1. Driving Factors of Water Yield Variation

The spatial-temporal pattern of water yield is resulted from comprehensive effects of multiple driving factors. In the past 20 years, climate change has profoundly influenced the variation of water yield in the HBOY Region, with a contribution rate of more than 80%. Precipitation and temperature are two crucial climate factors affecting water yield [67]. Precipitation has a direct positive impact on water yield since it forms the water input for basins, and temperature affects water yield through the process of evapotranspiration [55]. The contribution rate of climate change on water yield variation is a hot topic of water yield assessment [28,31]. By investigating the dynamic contribution rate of climate and land use change on water yield, we found that despite climate change is the dominant factor of water yield variation [27,68], the relative contribution rate of climate change is declining, while the contribution rate of land use change had a significant upward trend, which can be attributed to the increasingly intensified land use transformation under the process of vegetation restoration (GFG program) and urban expansion in the study area.

Land use layout affects several hydrological processes including evapotranspiration and confluence [69], thereby has an important impact on water yield. With rapid urban expansion, the area of impervious surface enlarges, which accelerates the formation of surface runoff (or underground runoff in places with drainage system) [3,70]. Furthermore, Redfern et al. found that urbanized surface had an effect of reducing evapotranspiration [71]. Therefore, built-up land has the highest amount of water yield, and the conversion from other land use types to built-up land would increase water yield. For the same reason, the water yield of barren land is relatively high, with a large proportion of rainfall directly forming surface runoff [46]. In contrast, well-vegetated land use types such as forest and grassland have relatively low water yield due to faster infiltration rate and strong transpiration of plants [64]. In the past 20 years, land use change in the HBOY region has mainly had a negative impact on water yield, mainly due to the conversion from cropland to grassland or forest under the GFG program, which increased vegetation transpiration since cropland generally has less canopy coverage than grassland and forest.

### 4.2. Policy Implications for Land Use Planning

Previous studies illustrated that scenario analysis could assist decision-makers in adopting an appropriate land use policy from the perspective of enhancing water yield [6,37].

This paper made progress in simulating water yield under multiple land use policy scenarios and future climate conditions. In the HBOY region, the RGD scenario (aiming at improving forest and grassland coverage) generates the lowest water yield, while the GSD scenario (aiming at retaining cropland area) has the highest water yield in SSP1-2.6, SSP2-4.5, SSP3-7.0, SSP5-8.5. This result indicates that despite the widely-known benefits of forest/grassland expansion [38,72], such as preventing soil erosion and improving water quality, there is a drawback of diminishing water yield which should be considered as a trade-off factor during the policy-making process, and afforestation should be implemented within the range of water resource carrying capacity. In the GSD scenario, enhancing water yield could be realized in synergy with cropland protection, however a higher water yield in the GSD scenario isn't always a good outcome, since it will bring about more surface runoff, leading to risks of flood disaster and soil erosion [37,69], especially in future climate conditions with high precipitation. From a comprehensive decision-making perspective, cropland protection policy is recommended in low-precipitation climate conditions (e.g., SSP1-2.6) to enhance water yield service and provide more available water for socio-economic systems; while afforestation policy and Green-For-Grain policy are appropriate in high-precipitation climate conditions (e.g., SSP5-8.5) to reduce the risk of natural disasters (e.g., flood, soil erosion).

### 4.3. Uncertainties and Limitations

Any model has limitations and so does the InVEST model. There are inevitable differences between simulated water yield and observed water yield (Figure 11) due to a series of causes. A major factor leading to this uncertainty from the InVEST model is the empirical Z parameter [44]. In this study, we calibrated the Z parameter according to measured runoff data (including surface runoff, subsurface runoff and base flow). The root mean squared error (RMSE) between simulated annual water yield and measured annual water yield was 7.73 mm. Compared to other calibration methods such as using number of rain events [47] or globally available $\omega$ [73], this approach could minimize the deviation between simulated results with actual water yield. Another limitation of InVEST model is its simplified assumption of the confluence process, which fails to distinguish the amount of surface water and underground water [70]. However, since this study mainly considers the total amount of water yield rather than its constitution, the outcome of InVEST model provides sufficient information for analysis. In addition, focusing on variation trends and the difference between scenarios reduces the flaw of uncertainty in the absolute value of simulated water yield.

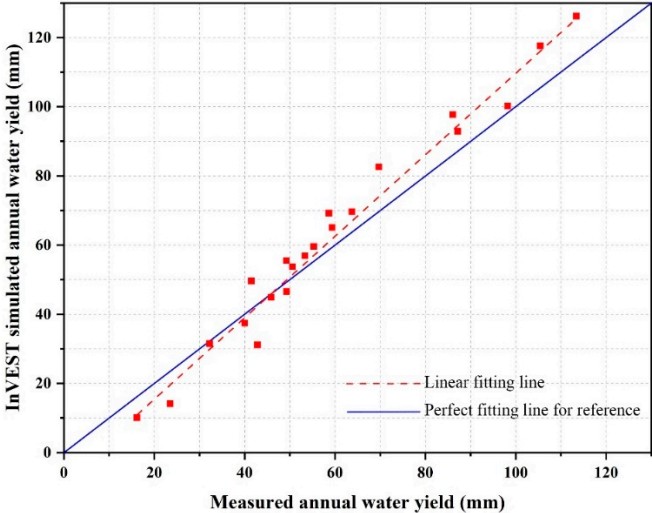

**Figure 11.** InVEST simulated annual water yield against measured annual water yield.

Limitations also exist in the policy scenario settings of this study since we used relatively radical assumptions (e.g., forbidding the transformation from forest/grassland to other lands) to distinguish the effect of different land policies. The extent of policy implementation in the real world has large uncertainty, and our projection results could only provide directional reference. Besides, there are many other policy tools [33,74,75] for coping with climate change, enhancing water yield, and mitigating water scarcity. Since we only focused on water yield rather than water reallocation, the effect of conservancy facilities and drainage systems wasn't incorporated in our analyzing process. Therefore, for future work the coupling effects of various policies on water yield deserve to be further studied; and we will consider the water reallocation effect of regional water conservancy facilities and drainage systems to better assist the decision-making process of water resource management.

## 5. Conclusions

This study focuses on exploring the dynamic effects of climate and land use change on water yield in drylands, as well as the future water yield under different land use policy scenarios. Taking the HBOY region in the northwest of China as a case of dryland area, we applied the methods of InVEST model, trend analysis, comparative experiments, and PLUS model for realizing the above research purposes. The results showed that water yield in the study area fluctuated from 2000 to 2020, with an average annual water yield of 62.5 mm. There's 5.8% of the total area that showed a significant increasing trend of water yield and 0.2% that showed significant decreasing trend. By means of comparative experiment, we found that climate change played a dominant role in affecting water yield and its contribution rate ranged from 85.8% to 99.6%. The contribution rate of climate change showed a decreasing trend, while the contribution rate of land use change was on the rise. Land use change had a negative effect on water yield in the HBOY region during 2000 to 2020, mainly because of the transition from cropland to forest and grassland. For projecting water yield in the future under various land use policies, we simulated three land use patterns (ND scenario, RGD scenario and GSD scenario) using the PLUS model. In future climate conditions of SSP1-2.6, SSP2-4.5, SSP3-7.0, SSP5-8.5, the GSD land use policy scenario would generate the highest water yield and the RGD scenario has the lowest water yield. The results of this study could help to better understand the effects of climate change and land use policies on ecological water yield service, thus have enlightening significance for adjusting land use policies towards enhancing water yield in dryland regions.

**Author Contributions:** Conceptualization, L.A., S.Z. and L.S.; methodology, L.A.; formal analysis, L.A.; resources, L.S. and S.Z.; data curation, L.A.; writing—original draft preparation, L.A.; writing—review and editing, S.Z. and L.S.; visualization, L.A.; supervision, L.S.; project administration, S.Z.; funding acquisition, L.S. and S.Z. All authors have read and agreed to the published version of the manuscript.

**Funding:** This research was funded by Strategic Priority Research Program of the Chinese Academy of Sciences (No. XDA19040102), National Natural Science Foundation of China (No. 42071281), Project of China Geological Survey (No. DD20221828).

**Institutional Review Board Statement:** Not applicable.

**Informed Consent Statement:** Not applicable.

**Data Availability Statement:** Not applicable.

**Conflicts of Interest:** The authors declare no conflict of interest.

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
