# Peer review of "Dynamic Effects of Climate and Land Use Policies on Water Yield in Drylands—A Case Study in the Northwest of China"

_water, doi:10.3390/w14233940_

Round 1

Reviewer 1 Report

The manuscript is focused on the analysis of water yield in dryland region (Hohhot, Inner Mongolia in China) and its changes due to climate and land use factors. There are two models have been used for water yield calculations: InVest model and PLUS model. Calculations are based on the monthly climate data. For land use change the 3 policy scenarios have been used as the matrix among croplands, grasslands, and forest land. Calculations have been used for the period 2000-2020. Calculated results of water balance parameters have been analysed by basic statistical methods (well described). results have been clerly presented via trends and maps for individual years. Main conclusion: the precipitation regime has the most significant role on water yield (more than 86 %).

General remarks/recommendations:

1. In the title of manuscript the climate change is mentioned, but analysed period is only 20 years (2000-2020) - it´s not enough for the term of climate change based on the IPCC definition. I recommend to use only ...effects of climate and land use policies ...

2. For the ET calculation the empirical Hargreaves equation have been used. I am not sure whether some empirical coefficients have been verified or not ?

3. In the methodological chapter I´m not identified the grid resolution

4. In Fig.10 the SSP scenarios are mentioned but without detailed description

I recommend the manuscript for publishing, but after title change and some minor changes based on my recommendations.

Reviewer 2 Report

The authors did a huge amount of work however the MS should be much improved.

The introduction needs to explain why InVEST model was selected and why the years 2000, 2020, and 2030 were selected. How are these years related to the policies?

The study area has large areas of urban but InVEST does not take into account artificial drainage. How does this affect the results? The MS stated that an integrated framework was used but the trend analysis is based on measured data. Why do authors call their method an integrated framework? Uncertainty analysis should be included. The risks of using assumptions should be clearer

Reviewer 3 Report

Dear Authors,

Many thanks for the opportunity to read your article entitled “Dynamic effects of climate change and land use policies on water yield in dryland regions --- a case study in the middle of China”. I greatly enjoyed reading your article and I think that you are addressing a timely and relevant topic. However, after reading your research, I had some major concerns with your research, which prevented me from recommending its publication. I think that this article should be reconsidered by the editors after some revisions. I will elaborate on my concerns below. This response letter is structured in line with the structure of your article. I hope that these comments will help you in improving the quality of your research.

I feel that the introduction misses several relevant elements. The authors do not position effectively their research in the extant scholarly debate. The authors should clearly position their paper in the scholarly literature, emphasizing how they are going to contribute to the scientific debate. Enhancing the positioning of the article should lead the authors to identifying the knowledge gaps affecting the scholarly debate. At the moment, it is not clear what is the area of scientific debate that the authors are going to address with their research. In the current version of the introduction, inadequate information is delivered to pinpoint the originality and the relevance of this research. Furthermore, water yield is not well-defined.

Literature review section is entirely missing from the manuscript. Authors need to add a separate literature review section to identify the research gap and the paper’s contribution to scholarship. In order to supplement the literature review please consider the following relevant studies:

"Impact of Participation in Groundwater Market on Farmland, Income, and Water Access: Evidence from Pakistan." Water 14(12): 1832.

"Extreme weather events risk to crop-production and the adaptation of innovative management strategies to mitigate the risk: A retrospective survey of rural Punjab, Pakistan." Technovation: 102255.

"The Competitiveness, Bargaining Power, and Contract Choice in Agricultural Water Markets in Pakistan: Implications for Price Discrimination and Environmental Sustainability." Frontiers in Environmental Science 10.

"Agricultural advisory and financial services; farm level access, outreach and impact in a mixed cropping district of Punjab, Pakistan." Land Use Policy 71: 249-260.

The methods and results section are clearly described and need minor language improvements.

The discussion merely presents an overview of the study findings, but it does not critically contextualize the study results in light of the extant scholarly debate. As a consequence, the authors are unable to provide us with thick and consistent information about how they are adding to the scholarly debate. Implications for theory and practice are limited. The authors do not provide adequate insights about how we can advance the scholarly knowledge about the particular issues being addressed.

Finally, yet importantly, the authors do not effectively elaborate on a relevant and inspiring agenda for further developments as well as the limitations of the study.

Round 2

Reviewer 3 Report

In response to my earlier comments, the authros have made significant changes to the manuscript. I recommend the publication of the article.